# Liquid–Solid Core-Shell Microcapsules of Calcium Carbonate Coated Emulsions and Liposomes

**Mark A. Bewernitz [1], Archana C. Lovett [2] and Laurie B. Gower [2,\*]**

1   Department of Biomedical Engineering, University of Florida, Gainesville, FL 32611, USA;
    bewernitzmark@gmail.com
2   Department of Materials Science and Engineering, University of Florida, Gainesville, FL 32611, USA;
    arch0618@gmail.com
\*   Correspondence: lgower@mse.ufl.edu



**Featured Application: Biodegradable Core-Shell Microcapsules for liquid soluble active agents.**

**Abstract:** Micron-sized core-shell particles consisting of a calcium carbonate ($CaCO_3$) mineral shell and a fluidic core were generated using a biomimetic approach, for the purpose of use as biodegradable microcapsules for release of active agents. Dinoflagellate cysts, unicellular organisms which deposit a protective hard mineral shell around their soft and fluidic cellular interior, served as our inspiration. Using the biomimetic polymer-induced liquid-precursor (PILP) mineralization process, calcium carbonate coatings were deposited on charged emulsion droplets and liposomes. Light microscopy, scanning electron microscopy, polarized light microscopy, X-ray diffraction, and confocal fluorescence microscopy were used to demonstrate that smooth $CaCO_3$ mineral coatings can be deposited onto the high curvature surfaces of emulsions and liposomes to yield micron-sized microcapsules for the effective entrapment of both hydrophobic and hydrophilic active agents. These biodegradable and biocompatible $CaCO_3$ microcapsules are novel systems for producing a powdered form of fluid-containing capsules for storage and transport of pharma/chemical agents. They may be used in lieu of, or in conjunction with, existing microcapsule delivery approaches, as well as providing a convenient foundation for which polymeric coatings could be further applied, allowing for more complex targeting and/or chemical-release control.

**Keywords:** microcapsules; biodegradable particles; PILP process; liposome coating; emulsion coating; biomimetic processing

## 1. Introduction

Emulsions and liposomes are promising components of microcapsule release agent delivery vectors due to their tailorable surface properties through surfactant selection and/or the addition of additives, and to their ability to store large amounts of release agent in compartments separated from the bulk solution [1,2]. Oil-in-water emulsions can store hydrophobic release agents internally without altering the bulk solution properties. The dispersion is maintained for long periods of time due to the addition of surfactants at the interface which reduce surface tension between the two liquid phases. These dispersions allow for the storage, transport, and delivery of an active agent or drug loaded in the minor phase which would normally be poorly soluble in the bulk solution phase. Liposomes are entities comprised of a phospholipid bilayer (or multiple layers) enclosing an aqueous core. Hydrophilic release agents may be stored in the aqueous interior and layers, while hydrophobic agents may be sequestered in the lipid membranes surrounding these layers. This allows for the dispersion of active agents with differing chemistries within one particle system. Liposomes range in

size from 1 to 100 nm in diameter, making them generally smaller and more stable than emulsions. The phospholipid bilayer may be functionalized by incorporating lipids with headgroup-containing molecules and ligands designed for cell-targeted drug delivery applications [3–6].

Although these systems have shown great promise for release agent delivery in both industrial and biomedical applications, there are challenges associated with good manufacturing production (GMP) of the more complex systems due to the chemical instability or denaturation of the encapsulated compound in the manufacturing process, as well long-term stability issues since these nano-sized vectors tend to degrade and destabilize over time [7,8]. Emulsions in storage destabilize through flocculation, creaming, breaking, and Ostwald ripening over time. Liposomes in storage destabilize through aggregation and coalescence [9]. Using extensive emulsifiers to stabilize the emulsion may not be a suitable solution due to environmental and health concerns [10]. Liposomal delivery vectors are prone to degradation when experiencing changes in the bulk solution, such as changes in salinity, pH and temperature. In biological systems, the system may experience multiple destabilizing changes in all these parameters, causing further problems for regulatory testing and application.

The challenges of destabilization and premature leakage are often addressed through the encapsulation of emulsions and liposomes with a protective, bioactive/biocompatible shell, forming core-shell microcapsules. This approach is currently used in a variety of release agent delivery applications, such as in enhanced oil recovery [11], cosmetic agent delivery [10], food/flavor technology [12], drug delivery on a tissue/cellular scale [13] and in building materials [14]. The shell component of the microcapsule may serve to merely protect the delivery vector by partitioning the active agent from the bulk solution or may be a component of a complex microcapsule with surface functionalization directing the targeting or release profile of the internal active agent.

A common method to synthesize emulsion core-shell microcapsules while addressing stability and emulsifier concerns is through the Pickering process [15]. This process utilizes nanoparticles rather than surfactants at the water–oil interface of emulsion droplets to reduce surface tension, resulting in a hard-granular shell surrounding a liquid, release agent-containing core. Often this is cheaper and safer than relying solely on additional surfactants to stabilize the vector due to the expense, health, and environmental concerns of excess surfactant in solution [16]. Emulsion-derived core-shell microcapsules made through the Pickering process are used to deliver active agents in a variety of applications including cosmetics [10,17], food additives [18], and oil recovery applications [11], wastewater treatment [19], and as enhancements to building materials [20]. Core-shell particles derived through the Pickering process, however, have some limitations. The process does not create core-shell particles with a continuous protective shell but rather creates a loose collection of nanoparticles at the interface that adhere due to interactions between oil/water (O/W) interfaces and the surface properties of the nanoparticles. This can lead to core-shell particles that experience premature leakage of the entrapped active agent and are susceptible to degradation due to shear stress and to changes in the bulk solution environment. There is a current research focus on methods and formulations involving crosslinking at the interface and additional coating of the Pickering emulsion to address leakage and stability limitations [21,22]. These promising proposed methods may prove to be complex and expensive, offsetting the simplicity and cost benefits of using Pickering core-shell microcapsules [21]. It would prove beneficial to the field to have an additional method, at their disposal, which may be used in addition to, or in lieu of, the traditional Pickering process to address stability and leakage challenges without the need for other complex post-processing methods.

Liposomal delivery vectors are stabilized and functionalized through the addition of additives either onto or into the phospholipid bilayer to form a core-shell microcapsule. Current research is focused on understanding and addressing the challenges of destabilizing mechanisms of flocculation and coalescence of liposomal microparticles both in vivo and ex vivo [23]. Some common techniques involve the addition of additives, such as polyethylene glycol (PEG), polysaccharides [24,25], chitosan [26], and cholesterol [27], which yield liposomal core-shell microcapsules that are resistant to destabilization during storage and are functionalized for drug/agent delivery and uptake. These

current techniques have challenges and limitations including high processing cost, difficulty in process scale-up, detrimental effects on the drug/chemical release profile, and the loss of core-shell stability in shear stress environments [27–29]. The stability of liposomal core-shell drug delivery vectors under shear stress is of particular concern in delivery systems using inhalers where premature destruction or leakage of the drug/chemical occurs during atomization, nebulization, and inhalation [30,31]. It may benefit the field to have an additional tool at its disposal; a method to further stabilize liposomal microcapsules during lyophilization and storage and which may reduce leakage by application of a continuous shell which is applied and removed under benign conditions.

Emulsion and liposomal derived core-shell delivery vectors suffer degradation upon lyophilization making long term storage and transport difficult [32,33]. Without water that has created the self-assembly of these vesicles, the structure of the vesicles are often altered or collapses, leading to premature drug leakage and aggregation or fusion of the vesicles [32,34]. This is a particular challenge with liposomal core-shell drug delivery vectors which need to be in a solid dosage form that can be variably reconstituted and dosed at a later date depending on which route the drug is to be administered [33].

To address these limitations for the wide variety of emulsion and liposome-based delivery applications, we have developed an easily applied method of creating core-shell microcapsules where a liquid emulsion or liposome droplet is encapsulated with a mineral shell, which may offer increased stability and reduced leakage due to the continuity of the smooth and uniformly coated shell. The Dinoflagellate cyst is an inspiration for this method [35]. This single-celled organism coats itself with a smooth, continuous $CaCO_3$ biomineral shell, in effect making it a little living microcapsule. To mimic this ability to encapsulate a fluidic interior with a mineral shell, we use a biomimetic process called the polymer-induced liquid precursor (PILP) process. The PILP process reproduces many of the non-equilibrium crystal morphologies found in nature by precipitating the mineral in the presence of negatively-charged polypeptides which are similar to the acidic aspartic acid-rich proteins utilized by organisms to direct biomineralization in nature [36]. The negatively-charged polyelectrolyte sequesters calcium and carbonate/bicarbonate ions which induces or stabilizes [37] a liquid–liquid phase separation in the crystallizing media, yielding ion-enriched droplets on the scale of tens of nanometers or more in diameter. These PILP droplets deposit onto substrates, coalesce into a continuous coating, solidify and crystallize into calcium carbonate mineral through the expulsion of water and the polymer additive [38]. The presence of $Mg^{2+}$ ions during the formation of the $CaCO_3$ PILP precursor (at concentrations similar to sea water) can aid the polymer in producing the fluidic character of PILP droplets, while tending to reduce any side products of conventional rhombic calcite crystals [39]. The liquid precursor, when aided by these additional additives (such as $Mg^{2+}$) solidifies into smooth, continuous $CaCO_3$ films with preferential deposition on hydrophilic surfaces, as has been shown in previous work with mineral deposition on glass substrates [40] and under Langmuir monolayers [41]. These films of $CaCO_3$ can also be patterned using templates of self-assembled monolayers [42], where the PILP droplets were found to preferentially adsorb to negatively-charged carboxylate-terminated alkane thiols and to avoid adsorption onto the more hydrophobic methyl-terminated alkane thiols [41].

Most $CaCO_3$-based biominerals exhibit non-equilibrium morphologies (non-faceted crystals), many of which exhibit elaborate crystal morphologies with smoothly curved surfaces (e.g., sea urchin spines), which is energetically unfavorable via the classical crystallization. Our group has proposed that these features could be a result of a PILP-type process being involved in the formation of biominerals [36]. Thus, we thought this process might enable the formation of smooth mineral coatings upon charged surfaces of emulsions and liposomes, even though these meso- to microscale particles have a high degree of curvature, thereby enabling us to emulate the $CaCO_3$ shell of the dinoflagellate cyst [35]. Building upon previous work that utilized the polymer-induced liquid precursor (PILP) process [43], we now demonstrate the ability to form a smooth and continuous $CaCO_3$ shell around the high-curvature surface of emulsion droplets and liposomes, both of which contained model compounds to represent hydrophobic and hydrophilic release agents, to yield $CaCO_3$-coated microcapsules.

## 2. Materials and Methods

### 2.1. Emulsion Preparation

Oil-in-water emulsion droplets were prepared by blending, in a household blender, *n*-dodecane oil (Fluka, St. Gallen, Switzerland) containing 1% *w/v* stearic acid (Aldrich, St. Louis, MO, USA) and deionized water to form an emulsion with a 1:9 oil/water volumetric ratio. The deionized water was adjusted to the desired pH between 7 and 11 using 0.01 M NaOH (Fisher Scientific, Hampton, NH, USA) in deionized water prior to blending. For experiments requiring the entrapment of a hydrophobic model compound, Nile Red dye (Sigma, St. Louis, MO, USA) was added to the oil phase prior to blending.

### 2.2. Liposome Preparation

Unilamellar liposomes with diameters of < 2 μm were prepared using the solvent-mediated dispersion method [2]. 100 μL of an organic phase, consisting of 20 mg/mL 1,2-distearoyl-sn-glycero-3-phosphotidyl choline (DSPC) (Avanti, Alabaster, AL, USA) and 8 mg/mL cholesterol (Sigma-Aldrich, St. Louis, MO, United States) dissolved in chloroform, was injected at a steady rate of 0.25 mL/min into 3 mL of water. The water was at a temperature of 80 °C during the injections. The aqueous solution was then cooled for 30 min at 4 °C, resulting in the formation of large unilamellar liposomes with diameters up to 2 μM.

### 2.3. Core-Shell Microcapsule Synthesis

We pipetted 1 mL of either the emulsion or the liposomes into a 35 mm Falcon polystyrene Petri dish, followed by 1 mL of 80 mM/400 mM $CaCl_2/MgCl_2$ solution (Sigma-Aldrich, St. Louis, MO, USA), freshly prepared using deionized water, and filtered through a 0.22 μm Millex® syringe filter (Sigma-Aldrich, St. Louis, MO, USA). Then, 36 μL of a freshly prepared and filtered 1 mg/mL polyelectrolyte solution was transferred to the Petri dish by micropipette. In the case of emulsion coating, poly-(L)-D,L-aspartic acid sodium salt (10,300 g/mol, Mw) (Sigma-Aldrich, St. Louis, MO, USA) was used as the PILP process directing agent. In the case of liposome coating, polyaspartic acid sodium salt (14,000 g/mol, monodisperse, Alamanda Polymers, Huntsville, AL, USA) was used. The Petri dish was then covered by Parafilm™ (Bemis Company, Neenah, WI, USA), through which a small hole was punched, into which the outflow end of the tubing from an ultra-low flow peristaltic pump (Fisher Scientific, Hampton, NH, USA) was inserted. At a rate of approximately 0.032 mL/min, 2 mL of a freshly prepared and filtered 300 mM $(NH_4)_2CO_3$ (Sigma-Aldrich, St. Louis, MO, USA) solution was pumped into the Petri dish over a period of ~ 60 min. A schematic of the process and the desired product is shown in Figure 1. The resulting mineral product was collected and centrifuged at 8000 rpm for 10 min. The supernatant was discarded, and the product pellet was rinsed with anhydrous ethanol (Fisher Scientific, Hampton, NH, USA) and re-centrifuged under the same conditions. The second supernatant was discarded, and the product pellet was either resuspended in 2 mL of anhydrous ethanol or placed on a glass slide and dried under room temperature or vacuum conditions, depending on the characterization technique to be employed.

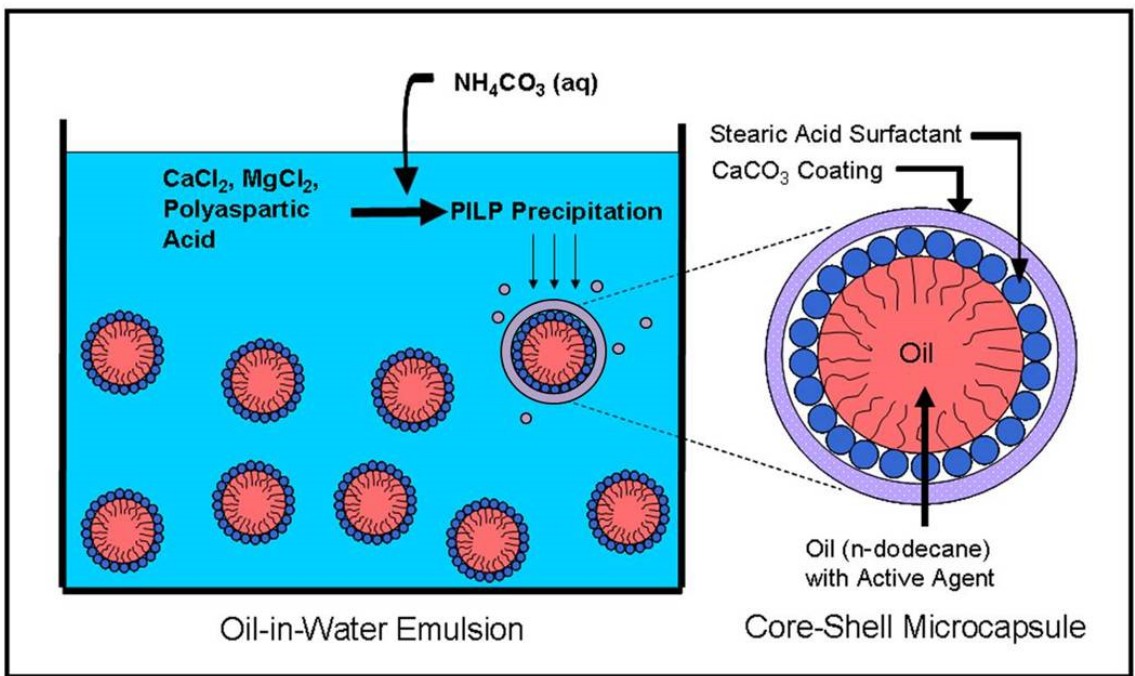

**Figure 1.** Schematic describing the synthesis of CaCO$_3$ core-shell microcapsules. Emulsion droplets or liposomes (emulsion shown above) are dispersed within an aqueous solution containing CaCl$_2$, MgCl$_2$, and polyaspartic acid. NH$_4$CO$_3$ (aq) is pumped into the solution to initiate the formation of polymer-induced liquid-precursor (PILP) droplets, which then adsorb onto the charged surface of the emulsion/liposome. The liquid or gel-like character of the PILP droplets allows them to coalesce into a smooth and continuous coating, which then solidifies and crystallizes into a continuous shell of calcite. An entrapped agent can be stored in either the aqueous interior of liposomes or the oily interior in emulsions (shown in figure).

## 2.4. Fluorescence Imaging of Encapsulated Model Compounds

To demonstrate the ability to entrap chemicals-of-interest in the microcapsules, Nile Red and rhodamine 110 (AnaSpec, Fremont, CA, USA) fluorescent dyes were used to simulate hydrophobic and hydrophilic model compounds, respectively. In the case of the emulsion-derived core-shell particles, Nile Red was added to the *n*-dodecane to achieve a fluorescent dye concentration of 20 μg/mL (~62.8 μM) prior to the emulsion synthesis. In the case of the liposome-derived core-shell particles, Nile Red was added to the chloroform organic phase to achieve a 20 μg/mL (62.8 μM) concentration, and rhodamine 110 was added to the water to achieve a 30 μg/mL (~81.8 μM) concentration prior to liposome synthesis. After synthesis, the microcapsules were centrifuged and washed as described in the core-shell microcapsule synthesis section and were stored in ethanol for imaging. Fluorescence imaging was conducted with a confocal microscope setup consisting of an Olympus IX-81 inverted microscope (Olympus Corporation, Tokyo, Japan) with an Olympus Fluoview 500 confocal scanning system (Olympus Corporation, Tokyo, Japan) with a tunable excitation laser. The images were taken with a 20 × 0.70 NA objective. The Nile Red was excited at 543 nm and the emission was collected using a 560 nm longpass filter after focusing the image to the highest fluorescence intensity. All images were taken and analyzed using the Fluoview software (Olympus Corporation, Tokyo, Japan).

## 2.5. Polarized Light Microscopy for Characterization of CaCO$_3$ Shell Crystallinity

Polarized light microscopy was used to detect the presence or absence of birefringence from the CaCO$_3$ shells to determine if the CaCO$_3$ mineral was amorphous or crystalline. The birefringence of core-shells created in a 2.5:1 ratio of Mg$^{2+}$/Ca$^{2+}$ environment (the CaCl$_2$ concentration described above was halved) were compared to core-shells created in a 5:1 ratio of Mg$^{2+}$/Ca$^{2+}$ environment. The light

microscopy images were obtained using an Olympus BX60 polarized light microscope (Olympus Corporation, Tokyo, Japan). The use of a 1st-order red gypsum λ-plate to display birefringence was used as the situation warranted, because it enables one to see both amorphous and crystalline materials simultaneously.

## 2.6. Scanning Electron Microscopy (SEM) for Morphological Analysis

To determine the overall particle morphology as well as the thickness of the mineral shell, microcapsules were created as described above and resuspended in 2 mL anhydrous ethanol. Then, 200 μL of the suspension was transferred to a 3″ × 1″ × 1 mm microscope slide (Fisher Scientific, Hampton, NH, USA) and allowed to air dry. A second microscope slide was placed on top of the microcapsules and gently tapped to crush the core-shell particles. The crushed particles were then sputter-coated with Au/Pd and analyzed using a JEOL 6400 SEM (scanning electron microscope) (JEOL USA, Peabody, MA, USA) with an accelerating voltage of 15 kV.

To demonstrate pH dependent degradation, core-shell microcapsules were created as described in the synthesis section except that, prior to final suspension in ethanol, the microcapsules were suspended in either 2 mL of 0.1 mM, pH 4, HCl aqueous solution or nano-pure water. In the case of the pH 4 HCl aqueous solution, the sample was resuspended and swirled gently, by hand, for 15 s at which point the solution was quenched with 2 mL of a 0.1 mN, pH 10 NaOH aqueous solution. Immediately following the quenching, 36 mL of anhydrous ethanol was added to halt any further degradation. The whole suspension was then centrifuged at 8000 rpm for 4 min, the supernatant was discarded, and the pellet was resuspended in 1 anhydrous ethanol for analysis. In the case of the pH 6.8 nano-pure water, the sample was resuspended and swirled gently, by hand, for 2 min. Immediately following the swirling, 18 mL of anhydrous ethanol was added. The whole suspension was centrifuged at 8000 rpm for 4 min, the supernatant discarded, and the pellet was resuspended in 1 mL anhydrous ethanol for analysis. Then, 200 μL of the degraded core-shells suspensions were deposited on a glass slide, dried, and sputter-coated with Au/Pd. The visual SEM analysis was conducted using a JEOL 6400 SEM (JEOL USA, Peabody, MA, USA) with a 15 kV accelerating voltage.

## 2.7. X-ray Diffraction (XRD)

To verify the crystalline phase of the $CaCO_3$ coating of the synthesized core-shell particles, X-ray diffraction (XRD) was conducted. Several batches of core-shell particles stored in ethanol were centrifuged at 8000 rpm for 10 min and the supernatant was removed. The centrifuge tube was then covered with parafilm and four small holes were punched into the parafilm with a sewing needle for the sample to dry overnight. The resulting powder was grounded and placed on a glass slide and then scanned with Cu-Kα X-ray radiation from a PANalytical X'Pert Pro Powder Diffractometer (Malvern Panalytical, Malvern, UK) at 45 kV and 40 mA, using a step size of 0.02° over a 2θ range of 10–150°.

## 3. Results

A $Mg^{2+}$-aided PILP process, as described in the methods section, was used to deposit smooth, continuous $CaCO_3$ shells onto the surface of oil-in-water emulsion droplets as well as aqueous liposomes. Light microscope images and polarized light microscopy (PLM) images of the core-shell microcapsules, created by PILP coating either oil-in-water emulsion droplets or liposomes, are shown in Figure 2A–D respectively. Coating the emulsion droplets resulted in products that were spherical $CaCO_3$ microcapsules with diameters of 2–10 μm, which is slightly larger than the original, uncoated emulsion droplets (Figure 2A). Core-shell microcapsules created by coating liposome vesicles using the PILP process appear to be much smaller, with a diameter of less than 2 μm (Figure 2C). This size is consistent with large liposomes, which are created by the ether-injection technique (~1 μm in diameter), that have been coated with a mineral shell.

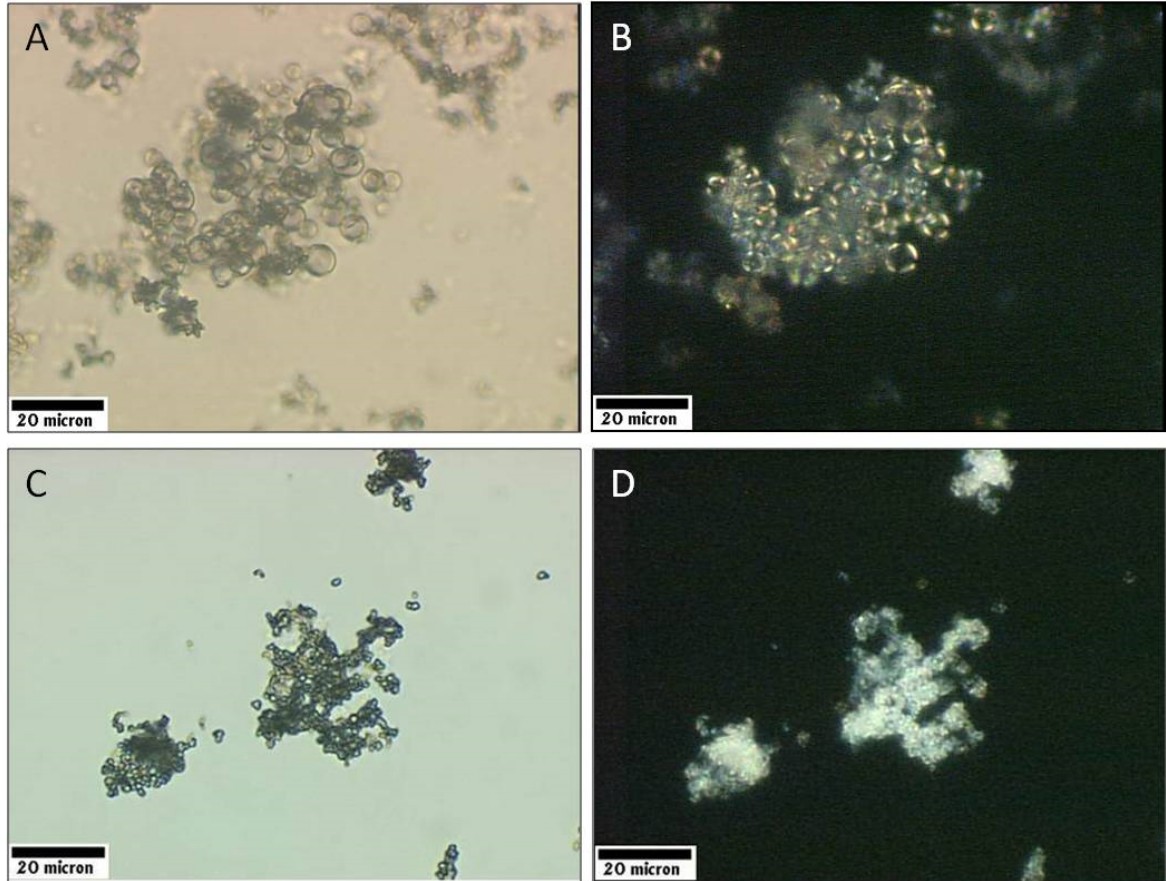

**Figure 2.** Light microscopy images of CaCO$_3$-coated emulsion droplets and liposomes. (**A**) Brightfield light microscopy of coated emulsion droplets. (**B**) Polarized light microscopy (PLM) of the same particles shown in (**A**). The visible portion (due to birefringence) displays a "Maltese Cross" pattern, indicating the product is crystalline with radial alignment of the polycrystals. (**C**) Brightfield light microscopy of coated liposomes. (**D**) PLM of the product seen in (**C**). Although very small, the clear interiors of the particles can be seen, suggesting a crystalline core-shell structure has been achieved.

As evidenced by the birefringent nature of the shells surrounding the emulsion droplets (Figure 2B) and liposomes (Figure 2D), the mineral shells appear to be crystalline. The presence of a "Maltese Cross" in the shell shown in Figure 2B, along with absence of birefringence in the center of the spheres, suggests a non-crystalline interior (presumably the n-dodecane fluid) with a surrounding shell composed of radially-oriented polycrystalline CaCO$_3$. Similarly, shells surrounding liposome-derived core-shells also display a birefringent coating surrounding a non-birefringent interior (Figure 2D), suggesting a crystalline core-shell structure. However, the presence of a "Maltese Cross" in the liposome-derived core-shells cannot be confirmed due to the resolution limit of the light microscopy technique.

SEM images of the mineral coated particles are shown in Figure 3. Notably, these originally fluidic particles have been dried for SEM analysis. Microcapsules generated by emulsion coating and liposome coating, shown in Figure 3A,B, respectively, display smooth, continuous coatings. There is some polydispersity in diameter, indicating that the coating process appears to coat emulsion and liposome particles with different diameters. Although not a focus of the current work, one could presumably make use of syringe filters or centrifugation to isolate core-shells of a desired monodispersed diameter. Additionally, since variable diameter emulsions and liposomes appear to coat well, methods to generate monodispered diameter emulsion or liposome substrates prior to coating may be a desirable approach to controlling the size of the core-shell product. A close-up image of a single emulsion microcapsule (Figure 4A) shows how smooth the surfaces of these particles are. Although there is a little bit of debris on the surface, we refer to this as smooth as compared to the alternative approach of nucleating crystals

on a Pickering template surface, which creates a surface composed of three-dimensional aggregates of rhombic calcite crystals [15]. The shell thickness of the coated emulsion microcapsules was determined by intentionally fracturing the microcapsules and obtaining SEM micrographs of the fractured shells. The thickness of the shells surrounding the emulsion-derived core-shells was found to be relatively uniform at ~ 600 nm for several batches of products all synthesized using the identical synthesis method described in the experimental methods section (Figure 4B).

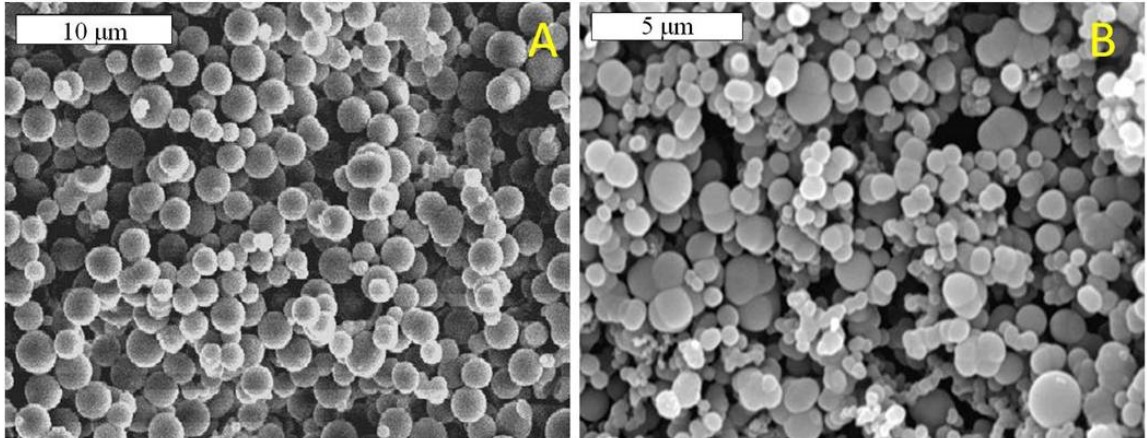

**Figure 3.** Scanning electron microscopy (SEM) images of the core-shell microcapsules. (**A**) Core-shell microcapsules with oil interior created by coating oil-in-water emulsion droplets with $CaCO_3$ using the PILP process. Their diameters range from 2 to 4 μm in this batch. (**B**) Core-shell microcapsules with aqueous interior created by coating liposome vesicles with $CaCO_3$ using the PILP process. Their diameters are distributed below 2 μm. Both emulsion and liposome-derived microcapsules are slightly larger than the emulsion droplets and liposomes alone due to the ~500 nm thickness of the applied mineral coating.

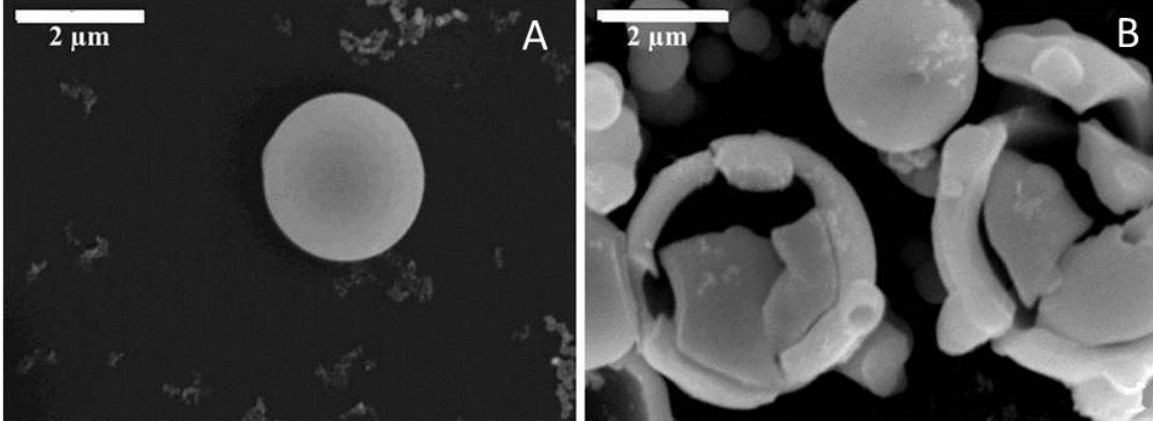

**Figure 4.** Scanning electron microscopy (SEM) images of intact and intentionally fractured $CaCO_3$-coated emulsion microcapsules. (**A**) The core-shell microcapsules consist of a smooth and continuous crystalline $CaCO_3$ coating. (**B**) Intentionally fractured microcapsules show a relatively thick continuous shell with a hollow interior where the liquid resided. The shell thickness was consistently ~500 nm using this procedure.

The polymorph phase of the $CaCO_3$ shell appears to depend on the conditions used during precipitation. As presented in Figure 5, an initial ratio of 2.5:1 $Mg^{2+}/Ca^{2+}$ in the mother solution yielded crystalline shells, but an initial ratio of 5:1 $Mg^{2+}/Ca^{2+}$ yielded an apparently amorphous mineral shell, even after gentle heating at 50 °C for 24 h.

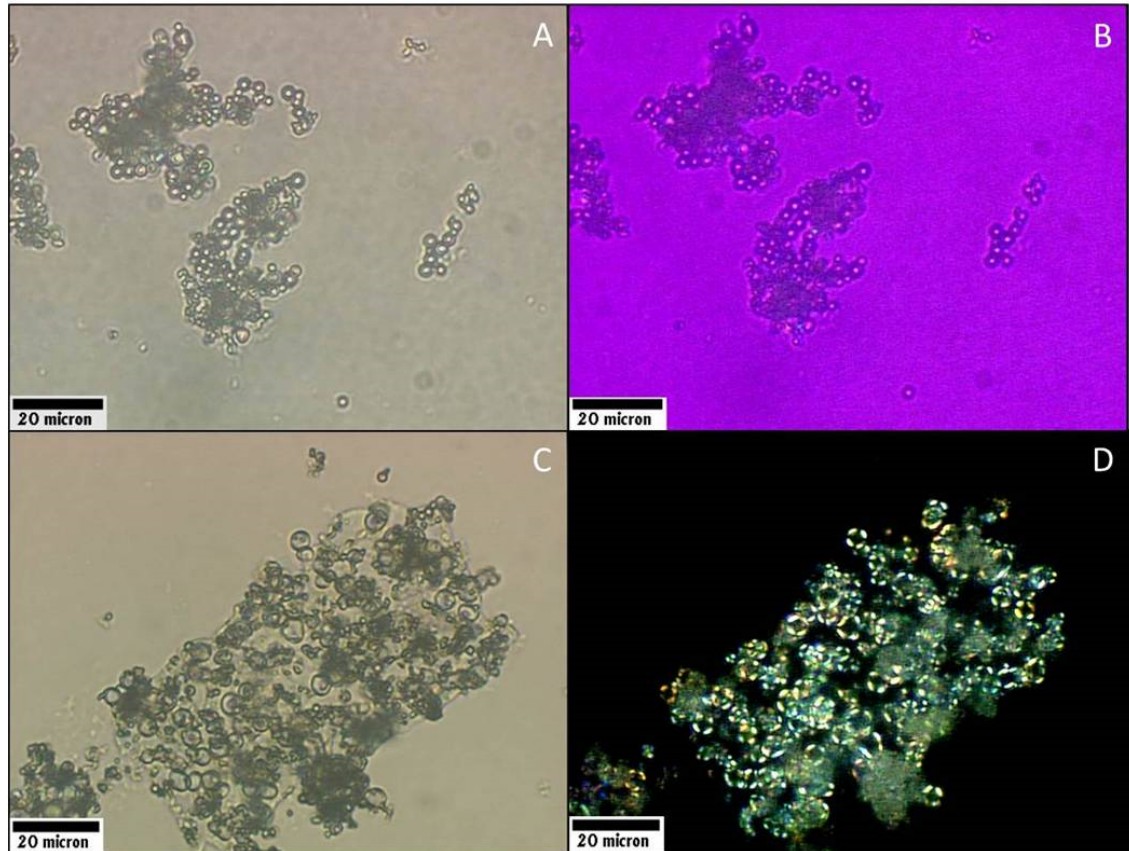

**Figure 5.** Tailoring the crystallinity of the microcapsules. The images on the left are optical micrographs, while the images on the right are using polarized light. (**A**,**B**) Microcapsules generated using a 2.5:1 ratio of Ca/Mg display birefringence, indicating crystallinity. (**C**,**D**) Microcapsules generated using a 5:1 Mg/Ca ratio lack birefringence, indicating that the mineral remains amorphous under these conditions. A gypsum waveplate was used for image B, giving a magenta (1st-order red) background, because the amorphous particles lack birefringence and are, therefore, difficult to see under crossed-polars. If they were birefringent, one would see higher order interference colors (such as orange and blue).

The amorphous core-shell microcapsules were smaller in size (2–3 μm) than the crystalline microcapsules (2–10 μm), which is presumably due to a difference in stability between the stabilized droplets, where the liposome system yields smaller metastable droplets as a template solid mineral shell deposition. These liposome-based microcapsules were not fractured as the larger emulsion-based microcapsules were due to their very small size. The thickness of the shell and the size of the liquid interior may be observed from the confocal micrograph shown in Figure 5.

Mineral phase identification of the $CaCO_3$-coating on liposome-derived microcapsules created using the 2.5:1 $Mg^{2+}/Ca^{2+}$ ratio was carried out using powder XRD (Figure 6). The pattern contains diffraction peaks that are consistent with the Joint Committee on Powder Diffraction Standards (JCPDS) standard for calcite, with some peak shifts caused by lattice strain from the incorporation of Mg into the lattice, which results in slightly smaller characteristic d-spacings. The results are consistent with the $Mg^{2+}$ incorporation behavior found in calcium carbonate films generated by the PILP process under similar $Mg^{2+}/Ca^{2+}$ ratios [39]. The absence of the characteristic signal of vaterite at 2θ of 25.0° and of aragonite at 45.9° suggest that neither of these phases is present in any significant amount. These characteristic peaks are indicative of polymorph presence and are used specifically to quantify the polymorph distribution in calcium carbonate samples [44].

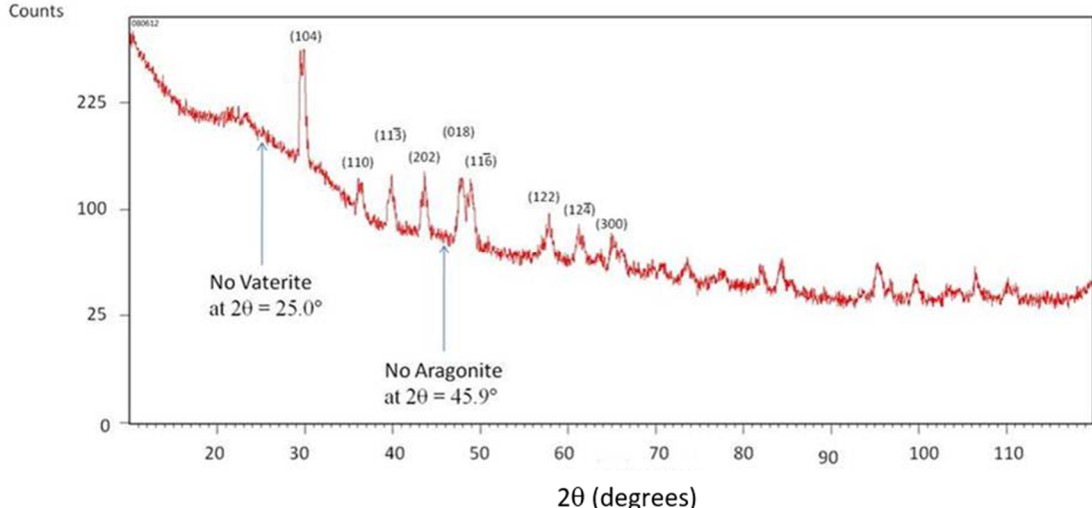

**Figure 6.** X-ray diffraction (XRD) analysis of the CaCO₃ shells of liposome-derived microcapsules. The XRD pattern display peaks characteristic of calcite. The theta values are slightly larger than expected for pure calcite due to magnesium ion incorporation (which is smaller than calcium ion, and thus reduces the lattice dimensions). The mineral coating does not contain significant amounts of vaterite or aragonite, as evidenced by the absence of peaks at their characteristic high intensity locations of 2θ = 25.0° and 45.9°, respectively.

Confocal fluorescence microscopy was used to demonstrate the internal entrapment of model active agent compounds within the core-shell particles. Capsules synthesized with rhodamine 110 and/or Nile Red fluorescent dye encapsulated in the interior were washed and subsequently examined. Figure 7 shows the entrapment of Nile Red fluorescent dye within the oil-interior of the emulsion-derived microcapsules. The liposome-derived core-shell microcapsules display the ability to entrap both hydrophobic (Nile Red dye) and hydrophilic (rhodamine 110) compounds, as is shown in Figure 8A,B, respectively, as well as in the overlay of both images shown in Figure 8C

Due to concerns that the CaCO₃ shells would degrade too slowly given that calcite has a rather low solubility and is notorious for clogging up industrial pipelines, etc., a preliminary study of the degradation potential of these particles was performed, the results of which are shown in Figure 9. The emulsion-derived microcapsules degrade readily in a pH-dependent manner, enabling a triggered release of active agent. The microcapsules exposed to the acidic (pH 4) environment for 15 *s* experienced severe degradation of the CaCO₃ shell in the form of large gaping holes. The microcapsules exposed to pure water (pH = 6.8) for 2 min experienced degradation as well, but the dissolution of the mineral was unevenly distributed around the microcapsule, and seemed to preferentially etch the "grain boundaries" of what appears to be remnants of the coalesced precursor particles. A remnant colloidal texture has been observed in PILP formed films before (as well as in biominerals) [42,45]. The overall amount of shell degradation appears to be similar for both conditions, but it is important to note that the microcapsules exposed to neutral pH required 8 times the exposure time, and that the degradation in pH 4 conditions appears to be a uniform thinning, whereas at pH 6.8 it exhibited a more asymmetric degradation.

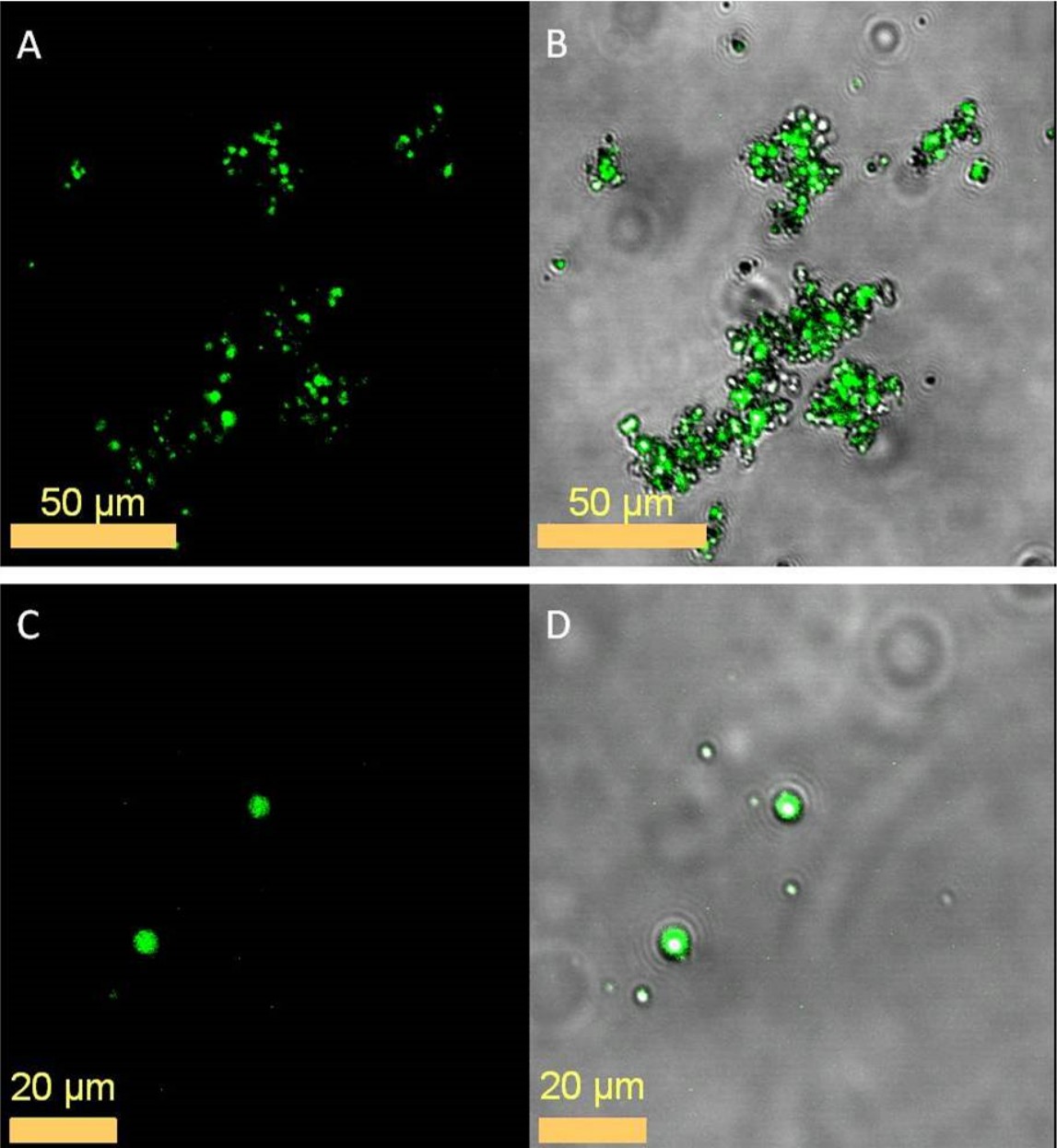

**Figure 7.** Confocal fluorescence microscopy of microcapsules from CaCO$_3$-coated emulsion droplets with Nile-Red fluorescent dye entrapped within the oily interior of the emulsion. Confocal fluorescence images are shown to the left (**A**,**C**), while the right images (**B**,**D**) show an overlay with the bright-field images. The high intensity of the fluorescence suggests that the entrapment of the active agent was very successful, as seen in both a large collection of particles in (**A**), as well as in individual microcapsules shown at higher magnification in (**C**). The fluorescence corresponds to the interior of the microcapsules for the cluster (**B**) and the individual microcapsules (**D**) when overlapped with their respective bright-field images.

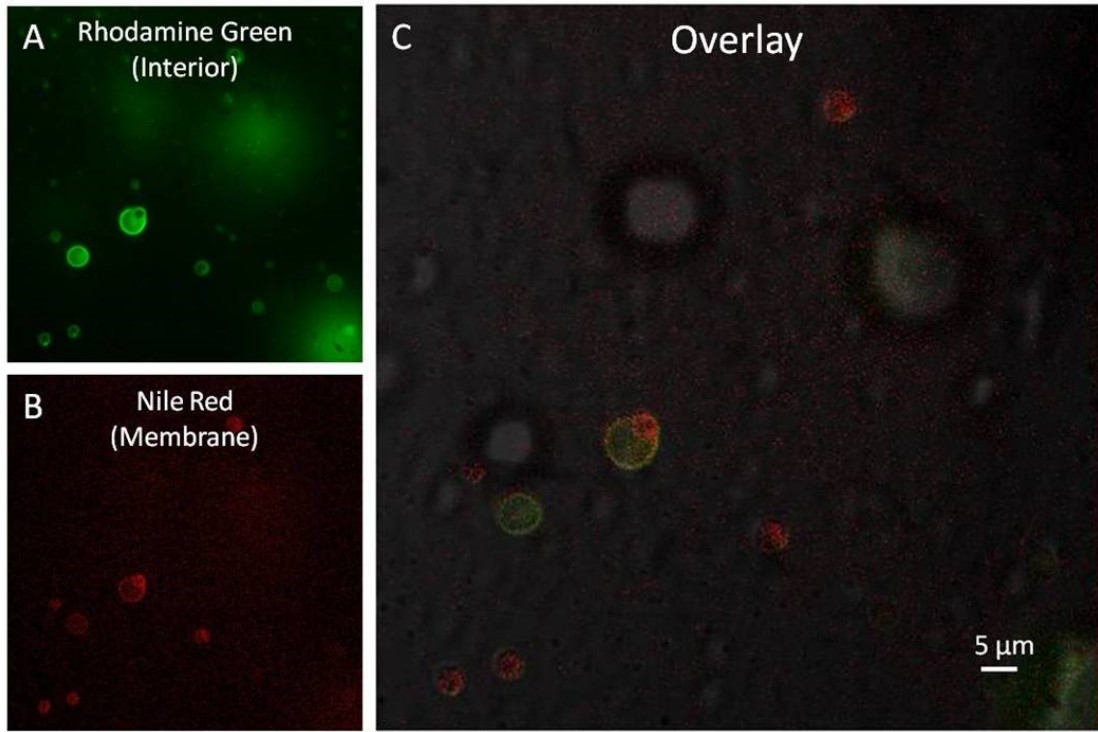

**Figure 8.** Confocal fluorescence microscopy demonstrating the entrapment capability of the $CaCO_3$-coated liposomes. Rhodamine 110 and Nile Red dyes were used as model hydrophilic and hydrophobic entrapment agents, respectively. (**A**) Rhodamine 110 is a hydrophilic fluorescent dye that would be expected to reside in the water interior of the microcapsule. The bright ring around the particles suggests that some of the dye may also bind to the charged headgroup of the liposome or be incorporated into the shell during the PILP process. (**B**) Nile Red is a hydrophobic fluorescent dye that would be expected to reside in the hydrophobic region containing the hydrocarbon tail-groups of the phospholipid bilayer. (**C**) An overlay of A and B shows that the liposome-based microcapsule system displays versatility in the types of encapsulated agents that can be incorporated.

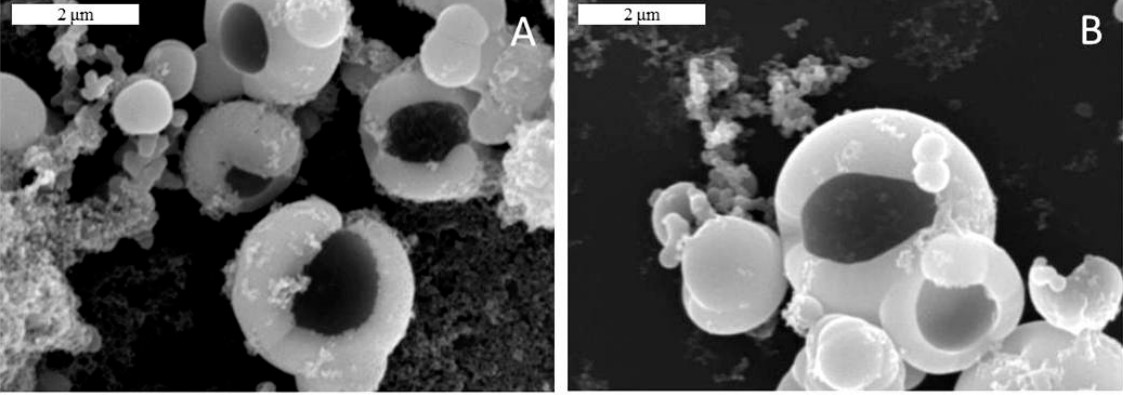

**Figure 9.** *Cont.*

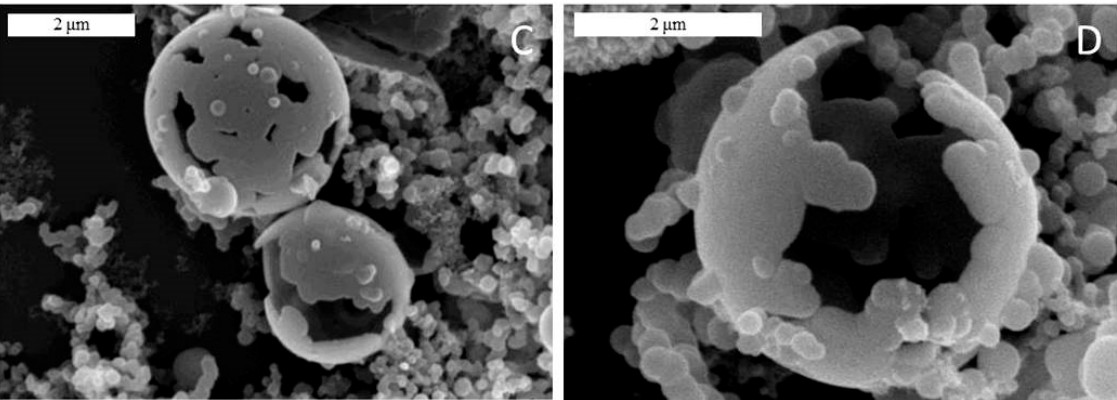

**Figure 9.** Particle degradation study. (**A**,**B**) Scanning electron micrographs of emulsion-based core-shell microcapsules after being exposed to a pH 6.8 nanopure water for 2 min, and (**C**,**D**) after exposure to a pH 4.0 HCl solution for 15 s. The rounded and particulate character of the remaining shells may be showing a memory effect of the PILP droplets that had initially coalesced to form the shells.

## 4. Discussion

The morphologies observed by optical and scanning electron microscopy (SEM) demonstrate that this biomimetic PILP process leads to a smooth, and continuous mineral shell that has formed a thick coating around the liquid-like particles. Importantly, SEM also demonstrates the robustness of core–shell particles when subjected to storage in ethanol, centrifugation, and air-dry storage. This suggests that these core-shell suspensions may be dried down to a powder and retain core-shell integrity. Using polarized light microscopy (PLM), we demonstrate the possibility of controlling the phase of the $CaCO_3$ shell (i.e., crystalline vs. amorphous). Control over the final phase could be a means to tailor the degradation properties of the microcapsules since amorphous calcium carbonate is metastable and far more soluble than the crystalline form. The core-shells' ability to entrap and protect active agents is demonstrated by the confocal microscopy results shown in Figures 7 and 8. This data allows one to discern between fluorescence originating from within the microcapsule and fluorescence originating from within the $CaCO_3$ shell or surrounding solution. Nile Red has the useful property of fluorescing only in very hydrophobic environments. Although it is conceivable that a small portion of the model compound could become entrapped within the mineral shell itself, the fluorescence seen in Figure 7 indicates that the interior of the microcapsules indeed contains the stored model agent. The results also show that the generated microcapsules retain the active agent even under the stress of centrifugation and resuspension in anhydrous ethanol. The presence and overlap of fluorescence originating from both oil soluble Nile Red and water-soluble rhodamine 110 (Figure 8C) demonstrates that the liposome-derived core-shell microcapsules can simultaneously entrap very different compounds in a single liposome-based microcapsule. Finally, the possibility of tailoring the degradation properties of these calcium carbonate coated core-shell microcapsules is also demonstrated by the altering of the polymorph of the calcium carbonate coating through adjustments in the magnesium-to-calcium ion ratio.

Much of the previous work developing calcium carbonate core-shell technology has focused on strategies to address loss of active agent activity while embedded within calcium carbonate [46], lack of affinity for calcium carbonate precipitation on/around active agent [47], and difficulties controlling the smoothness and morphology of the mineral coatings [48]. The work presented herein presents a means by which these challenges may be bypassed by coating an active agent-containing liquid droplet with a smooth, continuous calcium carbonate coating. A smooth calcium carbonate coating addresses premature agent release because there are no pores in the continuous shell. The mineral coating over an internal liquid reduces activity loss concerns because the active agent is stored in a favorable liquid environment during the benign (room temperature, aqueous based) coating process and during delivery.

Although free-standing flat films of CaCO$_3$ had been deposited on the anionic head groups of fatty acid monolayers in our previous studies [41,49], here we show that analogous surfaces containing the charged head groups of stearic acid and DSPC phospholipid are effective at templating the deposition of PILP precursors onto templating particles, even when there is significant curvature, which one might anticipate could create a large amount of strain on the mineral precursor, and particularly in the case of the tiny liposomes.

The particles consist of a "hard" mineral shell and "soft" fluidic oil core, in which a chemical of interest can be stored/protected at moderately high pH, in organic solvent, or dried down as a powder, until pH or mechanically triggered release. Although not examined here, one should be able to achieve the high encapsulation efficiencies analogous to what can be obtained by the core droplet emulsion and liposome systems to begin with, but with much easier separation and storage of the microcapsules once coated with a robust calcium carbonate shell. Storage and transport of a powder provides a major advantage over transport of the large amount of volume required in emulsions and liposome suspensions. The continuous coating allows even small molecular weight active agents to be stored in the interior of the core-shell without fear of early diffusion-release, and without having to design a new shell with different pore sizes. The non-faceted morphology of the mineral shells allows for easier deposition of further surface materials, such as polymeric coatings to further tailor core-shell functionality. The generated microcapsules retain the active agent even under the stress of centrifugation and suspension in anhydrous ethanol, suggesting they might readily allow for various chemical methods of surface modification. In addition, the liposome-derived microcapsules reported in this study demonstrate versatility in entrapment of either or both hydrophilic and hydrophobic compounds of interest.

Due to the versatility of the PILP-derived core-shell microcapsules described herein, we envision a wide-range of potential pharmaceutical, environmental, and industrial applications are possible. We anticipate that the release of the entrapped compound would be a relatively rapid or even a burst-type of release in the physiological environment. This might be useful for inhalant systems with dry powders that rapidly release the pharmaceutic agent into the moist environment of nasal or lung tissues [50] or for intragastric drug release systems that utilize the dissolution product of calcium carbonate to trigger pharmaceutical release and enhance pharmacokinetics [51]. Although not examined here, the benign processing conditions might also enable encapsulation of sensitive biological agents such as proteins, enzymes, growth factors, DNA, etc. Other applications might include environmental release of pesticide or fertilizer, or body care lotions that incorporate exfoliating particles, etc. These microcapsules may also be suitable for incorporation into composite materials where future rupture can release catalysts or other agents that lead to strengthening of the composite [52], and/or trigger self-healing in the material [53,54].

The core-shell particles shown in this study demonstrate that the PILP process can be used as a method coating emulsions and liposomes to enhance the properties of many different nano/microparticle systems. Stearic acid/n-dodecane (O/W) emulsion and 1,2-distearoyl-sn-glycero-3-phosphotidyl choline (DSPC)/cholesterol liposomes were chosen for use in this study because they are generally available and similar to many potential nano/microcapsule substrates in use. Due to the coating nature of this method, many of the resultant core-shell particle properties are largely substrate- and environment-dependent. The degree of polydispersity of the resultant core-shell microcapsules is dependent on the polydispersity of the emulsion or liposome substrate which are to be coated, for example. The synthesis method described in this report may lead to highly desirable encapsulation efficiencies in some systems but may require some adjustment to achieve desirability in other systems. A release profile of a model agent is not included in this report given the broad range of applications and thus environmental conditions that might be employed. The mineral shell deposited by the PILP process is proposed to stabilize existing delivery systems in storage and transport. The mineral shell, which is applied through relatively benign conditions, and can be removed under equally benign conditions, is expected to yield a functionalized nano/microcapsule with tailored release profile characteristics dependent on

the application. Although further research could functionalize the mineral shell to participate in a more complex release profile, that is beyond the scope of what is reported here. The Materials and Methods reported here provide a baseline method to which further studies may make modifications to tailor the core-shell particles to specific applications.

## 5. Conclusions

Core-shell microparticles were generated using the PILP process to coat oil-in-water emulsion droplets (for hydrophobic active agents) and phospholipid bilayer liposomes (for both hydrophobic and hydrophilic active agents) with a smooth and continuous calcium carbonate shell. The microcapsules may be suitable for potential use in drug delivery and other commercial applications where encapsulation of an active agent can be done within the fluidic interior of the core-shell system. The PILP method of generating microcapsules presented here may be used as an alternative to, or in conjunction with, current liposome- or emulsion-coating technologies to bypass or eliminate many of the challenges and limitations they face, such as leakage, premature release, destruction through lyophilization, storage instability, and ease of packaging and transport. The smooth and continuous calcite coating that the PILP process generates is quite remarkable considering that calcite usually forms faceted crystals when generated in vitro by the conventional crystallization process. Further investigations not included here may be required to tailor the core-shell microcapsule product depending on the intended application and its associated chemical environment. These further investigations include adjustments to the synthesis protocol to yield control over the size and size distribution, drug loading, release, and encapsulation efficiencies, as well as studies to evaluate the biocompatibility/bioactivity of the resulting calcium carbonate-coated microcapsule.

**Author Contributions:** Conceptualization, L.B.G.; methodology, L.B.G., M.A.B., A.C.L.; investigation, M.A.B. and A.C.L.; writing—original draft preparation, M.A.B.; writing—review and editing, L.B.G., M.A.B., and A.C.L.; supervision, L.B.G.; project administration, L.B.G.; funding acquisition, L.B.G. All authors have read and agreed to the published version of the manuscript.

**Funding:** This material is based upon work supported by the National Science Foundation under Grant No. DMR-0710605, and initial studies by the Particle Engineering Research Center (PERC) at the University of Florida, supported by the National Science Foundation (NSF) Grant No. EEC-9402989. Part of the research reported in this publication was supported by the National Institute of Diabetes and Digestive and Kidney Diseases (NIDDK) of the National Institutes of Health (NIH) under Award Number R01DK092311. The content is solely the responsibility of the authors and does not necessarily represent the official views of the National Institutes of Health.

**Acknowledgments:** We would also like to thank the staff at the Research Service Centers within the Herbert Wertheim College of Engineering for their training and guidance on the SEM and XRD instruments, and the staff at the McKnight Brain Institute for the training and assistance on the confocal fluorescence microscope.

**Conflicts of Interest:** The authors have no conflict of interest to declare.

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
