# Peer review of "Liquid–Solid Core-Shell Microcapsules of Calcium Carbonate Coated Emulsions and Liposomes"

_applsci, doi:10.3390/app10238551_

Round 1

Reviewer 1 Report

In this manuscript, authors have studied the development of functional core-shell microcapsules for drug delivery systems. This article describes how to develop a calcium carbonate coated emulsion droplets and liposomes by employing the mineralization strategy. Authors have shown a pH-responsive degradation mechanism; however, a set of experiments is required to confirm the bioactivity and biocompatibility of the developed nanocarriers in physiological conditions in vitro. Authors mentioned the importance of non-toxic carriers (biocompatible) for biomedical applications, and therefore it is beneficial to evaluate their produced microcapsules in this study. In my view, the manuscript (ID: applsci-994886) can be accepted by evaluating the biocompatibility and bioactivity of the microcapsules.    

- Study 1) model drug release profile, 2) cell viability, 3) cellular uptake and 4) intracellular drug release of the degradable microcapsules.

- Size distribution (e.g. polydispersity, aggregation, and agglomeration, etc.) of the final products (microcapsules) is needed to be evaluated.   

- Authors should explain how uniform-sized particles can be obtained (e.g. using syringe filters, centrifugation, etc.) and why these microcapsules have been produced in different sizes (diameters).

Reviewer 2 Report

The manuscript “Liquid-Solid Core-Shell Microcapsules of Core-shell microcapsules for the encapsulation of soluble active agents. In particular, the authors demonstrated the synthesis of calcium carbonate microcapsules using a particular methodology based on a bioinspired polymer-induced liquid-precursor mineralization process. In this process, the calcium carbonate was deposited over charged emulsion droplets and liposomes. The authors used microscopy techniques, X-ray diffraction, and confocal fluorescence microscopy to characterize the morphology and the properties of this new material. In addition, they studied the biodegradation of the microcapsules. Overall, the manuscript is well-written, and the data provided is potentially interesting for the field. I recommend the incorporation of additional data to complement the work before its publication in Applied Sciences.

Major and minor concerns:

1) Introduction Section. Please provide further information regarding the state of the art of previously described core-shell nanoparticle systems. Please discuss the main differences and advantages regarding previously reported methodologies and the system described here by the authors.

2) Results and Discussion section. The authors mention that the particles display “little polydispersity in diameter”. Further data regarding the size distribution of the microcapsules should be provided in the manuscript. In particular, DLS will provide a more accurate indication of the size distribution of the nanoparticles in solution. The associated PDI values need to be reported and discussed.

3) The encapsulation efficiency of the system should be evaluated.

4) Discussion section: The authors mention that “We anticipate that the release of the entrapped compound would be a relatively rapid or even a burst-type of release in the physiological environment”. Please provide experimental data to support this statement. A release profile is necessary to demonstrate the release behavior of the microcapsules.

Round 2

Reviewer 1 Report

Authors replied that their paper is not focused only on biomedical applications and drug delivery systems, but their introduction totally overviews biomedical applications and drug delivery systems, and needs to be comprehensively justified for other application such as energy and information technology and so on.

To stop confusing readers, authors should clarify in the results/discussion and conclusion that

  • Their products need modification in order to be narrow-sized.
  • The distribution of particles has not been studied and needs further investigations.
  • This product has not been tested in vitro and their bioactivity needs to be evaluated.
  • Drug loading capacity and release rate is unknown.
